# OpenReview forum: "Semantic Probabilistic Control of Language Models"
_ICLR.cc/2026/Conference — Submitted to ICLR 2026_

### Official Review · Reviewer_s9Dw · 2025-10-31

**Soundness:** 3
**Presentation:** 3
**Contribution:** 3
**Rating:** 8
**Confidence:** 3

**Summary:**

This paper proposes SConE, a method for exerting semantic control over the outputs of autoregressive language models at inference time. The approach leverages first-order gradient information from a sequence-level verifier (for attributes like toxicity, sentiment, or topic) to estimate, for each next-token choice, the probability that future generations will satisfy a given semantic constraint. This is achieved by constructing a locally contextualized approximate distribution and calculating the expected sentence embedding in that neighborhood. The resulting probability is used to reweight the LM’s next-token distribution. The authors evaluate SConE on tasks involving toxicity, sentiment, and topic control, reporting higher rates of constraint satisfaction compared to sampling-based baselines such as best-of-n, with little to no loss in fluency measured by perplexity. The method operates at inference time without requiring model fine-tuning and is designed to work alongside existing syntactic control techniques.

**Strengths:**

Originality: The paper presents a distinctive inference-time strategy for semantic control in language models, using verifier gradients to efficiently reweight generation probabilities, which differs appreciably from established sampling or heuristic approaches.

Quality: The theoretical justification is solid, and the core methodology is implemented and tested on well-chosen tasks. Experimental results demonstrate substantial improvements in constraint satisfaction without major disruptions to output quality.

Clarity: The writing is generally clear, figures are informative, and the organization facilitates understanding of both the motivation and technical details. Prior work is discussed sufficiently to highlight the contribution.

Significance: The work tackles an important and timely problem and shows practical effectiveness, indicating value and relevance to both researchers and practitioners interested in controlled text generation.

**Weaknesses:**

Comparative Experimental Coverage: while the experiments convincingly compare SConE to best-of-n and baseline generation methods, the paper does not empirically evaluate against methods in activation steering adaptors, or other types of steering.

Scalability and Efficiency Analysis: while SConE is presented as computationally attractive, a systematic analysis of its runtime and scaling characteristics is missing. The paper would be improved by benchmarking generation runtimes and memory use against baselines for several sequence lengths and constraint types.

Ablation and Sensitivity Studies: ablation experiments varying the parameters (e.g., different numbers of samples, alternative verifiers, different LM sizes) could provide insight into robustness and limitations, as well as guidance for practitioners on setting these hyperparameters.

Breadth of Task and Domain Evaluation: the current tasks rely on strong, well-curated verifiers. It remains unclear how SConE performs on less well-defined attributes or in more complex domains (e.g., multi-turn dialogue, factual correctness, style transfer). Commenting on this would be valuable.

**Questions:**

Please address weaknesses above.

---

> ### Author Response · Authors · 2025-11-23
>
> We would like to thank the reviewer for their feedback.
>
> > "Comparative Experimental Coverage: [...] does not empirically evaluate against methods in activation steering adaptors, or other types of steering."
>
> If we understand correctly the reviewer is requesting that steering methods are included in the experimental setup. We have already included LMSteer ([Han et al 2024](https://aclanthology.org/2024.acl-long.864/)), a lightweight steering approach that acts at the output embedding space, in our toxicity and sentiment experiments with GPT2 models (see Table 2 and Table 6), finding it to underperform SConE in both positive sentiment and detoxification control. We also include comparisons against DExperts, PPLM, and as of this rebuttal, FUDGE as well.
>
> > "Scalability and Efficiency Analysis: while SConE is presented as computationally attractive, a systematic analysis of its runtime and scaling characteristics is missing."
>
> We would like to thank the reviewer for this feedback. We refer the reviewer to the newly added section I of the appendix regarding the computation complexity of our approach. In summary
>
> All elements of our framework can be made independent of the model size**. Specifically:
> - The circuit scales linearly in the lookahead horizon as detailed below.
> - The lookahead samples are obtained by running a (small) masked LM on padded sequences.
> - Computing the attribute and it’s gradient require running a single forward/backward pass through the very computationally cheap verifier
> - Finally, computing the expected probability requires computing a single tensor product operation which can be computed efficiently using a single einsum operation
>
> We also added run times and peak memory consumption of ScoNE across different batch size, which we reproduce here
>
> | **Batch Size** | **1** | **2** | **4** | **6** | **8** |
> |----------------|-------|-------|-------|-------|-------|
> | **Avg. Time (s)** | 0.19 | 0.35 | 0.64 | 1.20 | 1.53 |
> | **Avg. Memory (GB)** | 8.85 | 13.71 | 20.91 | 37.86 | 41.38 |
>
> We also report relative run time differences as well as performance differences as a result of using different SCONE hyperparameters in Appendix C.
>
> > "Ablation and Sensitivity Studies: ablation experiments varying the parameters (e.g., different numbers of samples, alternative verifiers, different LM sizes)"
>
> In Appendix C, we report ablations over SCONE’s main hyperparameters, examining the effect of top-k (Appendix C.1) and the number of chains (Appendix C.2). While higher top-k improves constraint satisfaction, improvements plateau past top-k = 10 and come with an 8.71× increase in computational cost (to go from 10 to 25). Similarly, constraint satisfaction increases slowly as we increase the number of chains (and hence sample diversity), at a 5x fold decrease in computation.
> Moreover, we also validated the efficacy of our approach under different attribute verifiers in Appendix E.
>
> Based on the reviewer’s suggestion, we’ve conducted additional experiments to study the  generalization of SCONE to different model sizes of the Llama 3 family, which we report in Appendix D. Overall, we find similar results across model scales (less than 3% absolute points difference among model sizes), which suggests that SCONE is effective at steering regardless of model scale.
>
> > "Breadth of Task and Domain Evaluation: It remains unclear how SConE performs on less well-defined attributes or in more complex domains (e.g., multi-turn dialogue, factual correctness, style transfer). Commenting on this would be valuable."
>
> it is true that SConE’s performance is tied to the reliability of the verifier, and as long as a robust, differentiable verifier is available, one for which we can compute gradients, our method should be applicable irrespective of the task. This preposition is similar in nature to that of other methods (e.g., FUDGE, GeDI, DExperts), where control efficacy hinges on the quality of the verifier. Naturally, the noisier the signal, the more difficult it is to steer generations.
>
> One interesting avenue to consider is generative reward models with CoT (GenRM-CoT), that seem to outperform discriminative verifiers. The former are trickier to deal with in the context of our framework due to the non-differentiable sampling involved in CoT which in general makes it non-trivial to compute the gradient of the verifier w.r.t the input embeddings. One tempting avenue, however, is to apply any one of many gradient estimators at each sampling step to get an approximate gradient of the verifier w.r.t the input embedding. Other avenues are perhaps more involved and entail computing an approximate gradient using the method of finite differences, using a simultaneous perturbation stochastic approximation or polynomial approximations (e.g. Chebyshev approximations), among others. This is a very exciting avenue of future research. This would also open the door to factual correctness by querying a knowledge base, for instance.

---

### Official Review · Reviewer_aVGK · 2025-10-31

**Soundness:** 1
**Presentation:** 3
**Contribution:** 1
**Rating:** 2
**Confidence:** 5

**Summary:**

Work Summary
This paper focuses on the semantic control problem of language models (LMs), aiming to address the computational intractability of steering LM generations to satisfy sequence-level non-decomposable constraints (e.g., toxicity, sentiment, topic adherence). The authors propose a Semantic Control Estimator (SConE) that leverages verifier gradient information and knowledge compilation techniques: starting from a lookahead sample, a locally contextualized LM distribution favoring semantically similar sentences is constructed; the expected sentence embedding under this approximate distribution is computed via a decomposable and smooth circuit; finally, the next-token distribution is reweighted using the constraint probability estimated from the expected embedding and verifier gradients. The method is evaluated on three tasks (toxicity control, sentiment control, topic adherence control) with models like Llama-3.2 (1B) and GPT2-medium, showing that it outperforms baselines (e.g., BoN, DExperts) in constraint satisfaction while maintaining generation fluency and diversity.

**Strengths:**

Strengths
The experimental design is relatively comprehensive. The authors validate the proposed method across three distinct semantic control tasks (toxicity, sentiment, topic), covering both "constraint satisfaction" (e.g., detoxification, positive sentiment) and "constraint enhancement" (e.g., toxification) scenarios. For each task, multiple base models (Llama-3.2, GPT2-medium) and diverse baselines (training-free methods like random, beamsearch, BoN; training-based methods like PPLM, DExperts, LMSteer) are included for comparison. Evaluation metrics also cover multiple dimensions: constraint satisfaction (e.g., Toxic Prob, Sentiment Prob), generation fluency (perplexity), and diversity (Dist-1), providing a relatively holistic assessment of the method’s performance.

**Weaknesses:**

Critical Weaknesses and Reasons for Rejection
1.Severe Gaps in Related Work Comparison, Leading to Incomplete Contribution Positioning
The core idea of this work—Bayesian-based probabilistic inference for LM semantic control—falls into a research paradigm that was extensively explored around 3 years ago, yet the authors fail to compare with key representative works in this field, resulting in an unclear positioning of the work’s novelty. Specifically:
Omission of classic Bayesian control methods: Works like GeDi (Krause et al., 2021) and FUDGE (Yang & Klein, 2021) are foundational for Bayesian-based LM semantic control. GeDi uses generative discriminators to model attribute-aware distributions and adjust token probabilities via KL divergence constraints; FUDGE leverages factorized Bayesian inference to balance fluency and constraint satisfaction. Both share the same "probabilistic inference for distribution adjustment" core as SConE, but the paper does not discuss differences in technical routes (e.g., whether SConE’s "local contextualized distribution + knowledge compilation" is more efficient than GeDi’s discriminator-guided KL control) or comparative performance. Without this comparison, it is impossible to verify whether SConE truly addresses the limitations of earlier Bayesian methods or merely repeats existing ideas with minor tweaks.
Lack of discussion on paradigm evolution: The paper does not contextualize its method within the broader evolution of LM control research. Around 2021–2022, Bayesian-based methods (including the aforementioned GeDi, FUDGE) were widely studied, but with the advent of large language models (LLMs) like ChatGPT (2022), this paradigm has gradually been phased out. The authors ignore this key trend, failing to explain why a Bayesian-based method is still meaningful today, nor do they compare SConE with modern LM control approaches (e.g., instruction tuning, RLHF, in-context learning) that are more compatible with LLMs. This omission makes the work’s practical value questionable.
2. Ignorance of Critical Limitations of Bayesian Methods in Large-Scale LMs, Undermining Practicality
The paper claims that SConE maintains generation quality while improving constraint satisfaction, but it completely ignores a well-known flaw of Bayesian-based LM control methods: as LM scale increases, the trade-off between "constraint satisfaction" and "generation quality" becomes increasingly severe, which is precisely why this paradigm was abandoned in the LLM era. Specifically:
Scalability bottleneck: Bayesian methods like SConE rely on complex probabilistic inference (e.g., Gibbs sampling for lookahead samples, knowledge compilation for circuit construction) and gradient computations of verifiers. For small models (e.g., Llama-3.2 1B, GPT2-medium) used in the experiments, these computations are manageable, but for large-scale LLMs (e.g., GPT-4, Llama 3 70B), the computational overhead of inference and gradient calculation will increase exponentially. The paper does not evaluate SConE on large models, nor does it discuss optimization strategies for scalability, making the method impractical for real-world LLM applications.
Hidden quality degradation under strong constraints: The experiments only test "weak to moderate" constraint intensities (e.g., detoxification of non-toxic prompts, positive sentiment for movie reviews). However, in scenarios requiring strong semantic constraints (e.g., strict topic adherence for professional domains, zero-toxicity for child-friendly content), Bayesian methods often force the LM to prioritize constraint satisfaction by sacrificing lexical diversity and contextual coherence—manifested as repetitive expressions, rigid syntax, or disconnected logic. The paper does not design experiments for strong constraint scenarios, nor does it analyze potential quality degradation, leading to an overoptimistic assessment of the method’s performance.
Verifier dependency without mitigation: SConE’s constraint satisfaction is highly dependent on the accuracy of the sequence-level verifier (e.g., RoBERTa-based toxicity classifier). In the LLM era, verifiers often struggle to capture subtle semantic nuances of large models (e.g., sarcasm, implicit toxicity), and Bayesian methods amplify this dependency—errors in verifier judgments will directly distort the next-token distribution adjustment. The paper does not discuss strategies to mitigate verifier bias (e.g., multi-verifier ensemble, human-in-the-loop correction), further limiting the method’s robustness in practical use.
3. Insufficient Technical Justification for Core Modules
Key technical designs of SConE lack in-depth justification, raising doubts about the method’s reliability:
Arbitrariness of local contextualized distribution parameters: The paper uses "top-k=10" and "2 Gibbs sampling chains" as default parameters for constructing the local contextualized distribution, but does not explain why these parameters are chosen (e.g., why top-k=10 is better than top-k=5 or 25 in balancing efficiency and constraint coverage). The ablation study only briefly mentions that top-k=25 improves performance but increases time by 8.71x, without analyzing how parameter adjustments affect the "semantic similarity bias" of the local distribution—e.g., whether a larger top-k leads to diluted contextual information, or whether fewer sampling chains cause mode collapse.
Ambiguity in knowledge compilation implementation: The paper claims to use "decomposable and smooth circuits" to compute expected embeddings, but does not provide details on circuit construction (e.g., how to map the local contextualized distribution to circuit nodes, how to handle variable dependencies for long sequences). For complex sequences (e.g., 60-token topic control tasks), whether the circuit can maintain decomposability and smoothness, and whether the computational complexity remains tractable, are not verified. This ambiguity makes it difficult for other researchers to reproduce the work or assess its technical validity.

**Questions:**

same to weakness

---

> ### Author Response · Authors · 2025-11-23
> **response 1 to review**
>
> We would like to thank the reviewer for their feedback.
>
> > "Severe Gaps in Related Work”
>
> FUDGE trains classifiers on partial sequences to predict whether an attribute will be satisfied in the future, and uses Bayesian factorization to obtain the attribute-conditioned probability distribution. SCONE indeed uses a similar factorization but does not require learning any extra classifiers. GeDi on the other hand uses Bayes rule, but computes classification probabilities using the output of class-conditioned LMs rather than need to be training for each target attribute. This is not very feasible for current LLMs. Nonetheless we compared against DExperts, a very similar approach to GeDi that was shown to outperform it. This was also echoed in a more recent paper [1]. In our experiments, we show SCONE to outperform DExperts by a large margin across different domains. We have included an extended discussion comparing FUDGE, GeDi, and DExperts to SCONE when introducing our problem formulation in Section 3 as well as in the related work sections (Section 6 and Appendix F).
>
> We found it unnecessary to include GeDi as a baseline since we already include two baselines who were shown to be significantly superior to GeDI in both attribute control and fluency. These baselines are LMSteer [1] and DExperts [2]. In particular DExperts is very similar to GeDi where instead of class-conditioned LMs it leverages fine-tuned models for a particular attribute class. SConE outperforms both approaches.
>
> We have also added FUDGE as an additional baseline and show that SConE outperforms it by up to 16% points in toxification (Table 6) and by up to 40% points in sentiment steering (Table 2).
>
> > “Lack of discussion on paradigm evolution: […] nor do they compare SConE with modern LM control approaches (e.g., instruction tuning, RLHF, in-context learning”
>
> While instruction tuning, RLHF, and related alignment techniques have significantly improved the global behavior of large LMs, they do not provide per-instance, attribute-conditioned control. Moreover, these approaches are far less reliable for small and mid-scale models due to weaker instruction-following ability [3,4], and this gap becomes even more pronounced when multiple constraints must be composed simultaneously [1,2]. Real prompts frequently require several constraintsand existing benchmarks show that smaller models struggle to satisfy them jointly [5]. In addition, alignment methods require a separate fine-tuning run for each target attribute and for each model size [6], which is costly [9] and often infeasible in low-data regimes. In contrast, SConE provides fine-grained semantic control at decoding time, without additional data or model retraining, and works uniformly across scales.
>
> We also included two prompt-based baselines (Tables 1 and 6): a 5-shot prompting strategy and a fixed attribute prefix (AttrPrefix), following prior work [7,8]. **While 5-shot prompting performs similarly to SConE in the toxification setting**, it fails substantially in detoxification: 5-shot GPT2-medium produces at least one toxic continuation in 90.25% of cases compared to 1.50% for SConE. That is, **SConE achieves a 58× more reduction in toxic generations. in the detoxification setting**. AttrPrefix also fared a lot worse than SconE. We observed similar numbers for Llama (Table 1)
>
> [1] Bosi Wen, Pei Ke, Xiaotao Gu, Lindong Wu, Hao Huang, Jinfeng Zhou, Wenchuang Li, Binxin Hu, Wendy Gao, Jiaxin Xu, Yiming Liu, Jie Tang, Hongning Wang, and Minlie Huang. 2024. Benchmarking complex instruction-following with multiple constraints composition. In NeurIPS 2024.
>
> [2] Zhiyuan Zeng, Jiatong Yu, Tianyu Gao, Yu Meng, Tanya Goyal, Danqi Chen. 2024. Evaluating Large Language Models at Evaluating Instruction Following. In ICLR 2024.
>
> [3] Qwen3 Team. 2025. Qwen3 Technical Report
>
> [4] Llama Team, AI @ Meta. 2024. The Llama 3 Herd of Models
>
> [5] Jason Wei et al. 2022 Emergent Abilities of Large Language Models.
>
> [6] Yu Fei, Yasaman Razeghi, Sameer Singh. 2025. Nudging: Inference-time Alignment of LLMs via Guided Decoding
>
> [7] Jonathan Pei, Kevin Yang, and Dan Klein. 2023. PREADD: Prefix-Adaptive Decoding for Controlled Text Generation. In ACL 2023
>
> [8] Timo Schick, Sahana Udupa, Hinrich Schutze. 2021. Self-Diagnosis and Self-Debiasing: A Proposal for Reducing Corpus-Based Bias in NLP. In TACL 2021.
>
> [9] Pynadath, Zhang. Controlled LLM Decoding via Discrete Auto-regressive Biasing. ICLR 2025.
>
> > “The paper claims that SConE maintains generation quality”
>
> Our claims that SConE maintains generation quality are supported qualitatively through the examples provided in Appendix B (see Tables 7, 8 and 10) **and** quantitatively by measuring metrics related to diversity and fluency (see Tables 1, 2, 3, 6, and 9) across models of different model sizes (Llama 1B and GPT2-medium). We expanded the toxicity control experiments to assess SConE’s generalization to various model sizes (1B, 3B, 8B, and 70B) in Appendix D

---

> ### Author Response · Authors · 2025-11-23
> **response 2 to review**
>
> > "Scalability bottleneck:”
>
> Respectfully, it is not the case that the computational overhead of our approach scales exponentially with the model size. **In fact, all elements of our framework can be made independent of the model size**. Specifically:
> - The circuit scales linearly in the lookahead horizon as detailed below.
> - The lookahead samples are obtained by running a (small) masked LM on padded sequences.
> - Computing the attribute and it’s gradient require running a single forward/backward pass through the very computationally cheap verifier
> - Finally, computing the expected probability requires computing a single tensor product operation which can be computed efficiently using a single einsum operation
>
> Please see section I in the appendix for a full breakdown of the complexity.
>
> > “Hidden quality degradation under strong constraints”
>
> Respectfully, we disagree with the reviewer’s assessment on this aspect. Our experiments clearly demonstrate, quantitatively (Tables 1, 2, 3, 6 and 9) and qualitatively (in Appendix B Tables 7, 8, and 10), that SConE is able to achieve good constraint satisfaction without severe degradation of fluency or diversity of generations. In the specific case of strict topic adherence, we have shown in Table 3 that SConE is able to steer Llama (1B) towards satisfying the target topic ~95% of the time **without sacrificing either the perplexity or the diversity** of the model’s generations. It has also been shown, time and again, that we are able to enforce **strict logical constraints**, where one would arguably observe the large quality degradation, without sacrificing the perplexity or diversity of the generated text [3,4,5], to the extent that it was even implemented by OpenAI in ChatGPT [6].
>
> > “Verifier dependency without mitigation”
>
> The use of discriminative verifiers is not unique to our work. That being said, our approach is especially suited to dealing with less than accurate verifiers as it balances linguistic plausibility and attribute satisfaction by virtue being derived probabilistically from first principles: a continuation with moderate linguistic probability and moderate attribute satisfaction is picked over a continuation with low probability but very high attribute satisfaction. Therefore, attribute satisfaction does not blindly guide our generation.
>
> > “Arbitrariness of  parameters”
>
> SConE is, at its core, an estimator of semantic constraint probability (see Equations 3 and 4). Generally, most estimators allow for parameters that trade off compute, accuracy, and efficiency. This is also prominent in almost all works on LM control: GeDI, FUDGE, DEXPERTS, SMC, etc. We carried out a small hyperparameter search, which can be found in Appendix C. We identified a good set of hyperparameters that tradeoff accuracy and compute, and those are the hyperparameters used by default.
>
> The local contextualized distribution, and the quality thereof, is unaffected by the hyperparameters. Instead, given a reference sentence $y_{\text{ref}}$, for any other sentence $y$, its probability is computed by conditioning on $y_{\text{ref}}$. Rather, the hyperparameters affect the quality of SConE estimate by manipulating the **number** of such local distribution we average over.
>
> > "Ambiguity in knowledge compilation”
>
> We would also like to point out that we never need to “materialize” a circuit. Rather, we draw a parallel between how our proposed approach models the structure of a circuit, thereby providing tractability guarantees. In fact, computing the expected embedding reduces to performing a very efficient batched matrix multiplication. This computation scales linearly in T. We have updated Section 4.3 in the paper to make this clear.
>
> References:
>
> [1] Chi Han, Jialiang Xu, Manling Li, Yi Fung, Chenkai Sun, Nan Jiang, Tarek Abdelzaher, Heng Ji. Word Embeddings Are Steers for Language Models. In ACL 2024.
>
> [2] Alisa Liu, Maarten Sap, Ximing Lu, Swabha Swayamdipta, Chandra Bhagavatula, Noah A. Smith, and Yejin Choi. 2021. DExperts: Decoding-Time Controlled Text Generation with Experts and Anti-Experts. In ACL 2021
>
> [3] Kareem Ahmed, Kai-Wei Chang, and Guy Van den Broeck. Controllable generation via locally constrained resampling. In ICLR 2025.
>
> [4] Honghua Zhang, Po-Nien Kung, , Masahiro Yoshida, Nanyun Peng, and Guy Van den Broeck. Adaptable logical control for large language models. In NeurIPS, 2024.
>
> [5] Vincenzo Collura, Karim Tit, Laura Bussi, Eleonora Giunchiglia, and Maxime Cordy.
> ABS: Enforcing Constraint Satisfaction on Generated Sequences via Automata-Guided Beam Search. Arxiv Preprint.
>
> [6] https://platform.openai.com/docs/guides/structured-outputs

---

### Official Review · Reviewer_afpf · 2025-10-31

**Soundness:** 3
**Presentation:** 2
**Contribution:** 2
**Rating:** 4
**Confidence:** 4

**Summary:**

This paper proposes a way to control the semantics of LLM generations at inference time. The method (SConE) operates at decoding, reweighting the LLM's output token probability distribution in a way that privileges desired semantics of the future output. "Desired semantics", e.g., not being toxic, is given by a neural attribute classifier (e.g. a BERT-like toxicity classifier) that takes strings to ratings in [0,1]. The paper shows that the problem of reweighting the output token probabilities computationally boils down to finding the future generations' expected embedding under the attribute classifier. The expected embedding step of the approach is efficient because it can be vectorized. The process of sampling future generations can be made more efficient using HogWild! Gibbs sampling. Experimentally, the approach outperforms existing baselines.

**Strengths:**

1. ScoNE is lightweight method that demonstrably outperforms existing baselines on toxicity, sentiment, and topic control.
2. The method is timely and relevant, as it it proposes a way to control generation by LLMs using only their output probabilities.

**Weaknesses:**

A primary weakness of the paper is insufficient contextualization wrt prior work. A missing paper is FUDGE https://arxiv.org/pdf/2104.05218 which also operates by reweighing token probabilities with Bayes rule, via some attribute classifier on future generations. Since FUDGE is very close to your method, it should be discussed up front and tested as a baseline.

A second crucial weakness of the paper is the mathematical exposition. The paper was difficult to read, taking many close readings in order to grasp the intent of the paper flow. The main detractor was the abruptness with which the writing moved from concrete to very abstract topics (e.g. knowledge compilation). In general, the lack of "connective tissue" between sections put the burden on the reader to infer the author's intent. These however should be easy to improve.

If both points are addressed, I will increase my score to an accept.

## Detailed comments

1. l118 Needs some explanation here of Y_i, e.g., "...of random variable Y_i over the output token at time t=i."

2. Eq 2 is very similar in spirit to previous works such as FUDGE. Please add references here to at least FUDGE (maybe also DExperts, Gedi) and state the extent to which ScoNE is similar; if not, why not.

3. Section 4.1 I found this section to be confusing as it lacked concrete examples. What does it mean for "different sentences to depend on different sets of conditionals"? Can you edit this paragraph to map "sentences" and "conditionals" concretely onto the mathematical expressions in Eq (6)? Also, could you provide a concrete natural language example?

4. Similarly, I did not understand the difference between $y$ and $\tilde y$. Can you provide an example of a semantic neighborhood $\tilde y$ compared to $y$? I find the current explanation to be rather opaque and disconnected to later sections.

5. Algorithm 1: Please add the algorithm of CondMarginals somewhere

6. l294 "We denote by emb:..." the introduction of $\overline{emb}$ needs to be gentler, given the previous paragraphs. I would first write Eq (7) purely in terms of $s$ and $y_{1:T}$, then motivate the current Eq (7) using the fact that the embedding function of the verifier is a deterministic function of the LM generation-- i.e., we can do the Taylor trick replacing the token outputs with their embedding. For consistency, I would replace $\nabla \phi_a(s)$ with the version you have in Algo 3: $\nabla_{emb(s)} \phi_a$. Otherwise, it's not transparent as written; I had thought that you were taking the gradient with respect to $s$, thereby missing a chain rule from the embedding to $s$ in the first-order Taylor expansion.

7. l301 Do we have any theoretical guarantees on how good this approximation is?

8. l309 We need to better motivate the path from knowledge compilation to average embeddings here. These few paragraphs start by introducing decomposability and smoothness of circuits. I would instead start by stating the goal, i.e., to compute the expected embedding from samples using a single pass.

9. Relatedly, the discussion on how to compute the expected embedding (l343) reads as disconnected to the previous part. If I understand correctly, this part describes how to, given a collection of sampled outputs, do one pass through the computation graph of the BERT verifier and produce the expected embedding over this sample. This is not obvious given how it is written. I recommend that, instead of only introducing $\tilde p$, also stating that this circuit is the "computation graph" of the $emb$ function (i.e. the part of the verifier that you extract the embedding out of).

10. Finally, a connection is missing from the last part of that subsection to the circuit $\tilde p$ being smooth and decomposable. It doesn't need to be a formal proof, just a short justification that your verifier of choice has this property. Alternatively, because it takes up too much space and doesn't add much to the main contribution, I would even move the smoothness/decomposable parts entirely to the Appendix.

**Questions:**

Here are some more concrete suggestions to smooth the writing.

1. l269 to approximate ..., the constraint probability, *such as the likelihood a sentence is toxic*.

2. l408 State the sample size for ScoNE in the main (or mark clearly in Fig 3).

3. l193 Next, we will show how to compute *an approximation of* the above expectation in closed form...

4. Dist-1 is the type-token ratio (TTR), a well-known measure in NLP and linguistics. I would suggest renaming it for consistency with the literature.

---

> ### Author Response · Authors · 2025-11-23
> **Response 1 to review**
>
> We would sincerely like to thank the reviewer for engaging with the paper and for their detailed suggestions.
>
> > “FUDGE is very close to your method, it should be discussed up front and tested as a baseline.”
>
> We have included a discussion comparing FUDGE and GeDi, and DExperts to SCONE in the introduction, when introducing our formulation (in Section 3), as well as made sure to add them to the related works section in Section 6 and  Appendix F.
>
> In terms of methodology, FUDGE trains classifiers on partial sequences to predict whether an attribute will be satisfied in the future, and uses Bayesian factorization to obtain the attribute-conditioned probability distribution. SCONE indeed uses a similar factorization but does not require learning any extra classifiers. GeDi and DExperts on the other hand uses Bayes rule, but computes classification probabilities using the output of class-conditioned LMs/Experts that need to be trained for each target attribute. This is not very feasible for current LLMs. Please see our revised text in Section 3.
>
> Qualitatively, please see the updated tables 2 and 6 in the revised submission. In Summary, while FUDGE is comparable to SCONE on detoxification, achieving comparable perplexity and being slightly less diverse, **SCONE largely outperforms** FUDGE on toxification settings: **up to 16% absolute point difference on both toxicity metrics in the non-toxic subset**. Similar patterns are observed in sentiment control, where SCONE generates all positive continuations for about 79% of the prompts, whereas FUDGE only does so for 7% of the prompts. In fact, we observe that the average minimum sentiment across 10 generations is about 46.08 for FUDGE whereas SCONE’s minimum sentiment is about 83.98, a 1.82x increase over FUDGE.
>
> > "the lack of "connective tissue" between sections puts the burden on the reader to infer the author's intent.” ... “l118 Needs some explanation here of Y_i,”
>
> Please see the revised submission
>
> >“Eq 2 is very similar in spirit to previous works such as FUDGE. Please add references here to at least FUDGE (maybe also DExperts, Gedi) and state the extent to which ScoNE is similar; if not, why not.”
>
> We added references, methodological comparison, as well as quantitative comparison.
>
> > “What does it mean for "different sentences to depend on different sets of conditionals"? Can you edit this paragraph to map "sentences" and "conditionals" concretely onto the mathematical expressions in Eq (6)? Also, could you provide a concrete natural language example?” ... “Can you provide an example of a semantic neighborhood $\tilde{y}$ compared to $y$”
>
> It means that each candidate sentence requires evaluating a different set of conditionals. Mathematically, for any sentence $y$, this is denoted by the dependency of $\tilde{p}(y)$ on the conditionals $p(\cdot | \mathbf{y}_{-i})$.
>
> Concretely, given two sentences (“I like dogs”) and (“I like cats”), their pseudolikelihood would be
>
> $\tilde{p}(\text{I like dogs}) = p(\text{I}| - \text{like } \text{dogs}) \times p(\text{like}|\text{I } - \text{ dogs}) \times p(\text{dogs}| \text{I } \text{like } -)$
>
> $\tilde{p}(\text{I like cats}) = p(\text{I}| - \text{like } \text{cats}) \times p(\text{like}|\text{I } - \text{ cats}) \times p(\text{cats}| \text{I } \text{like } -)$
>
> whereby evaluating N candidate sentences requires $T \times N$ forward passes through the masked LM, one for each masked position in each sentence. The problem is not just cost, it is that pseudo-likelihood factors for different sentences are conditioned on different contexts, so their scores are neither comparable nor normalizable. By fixing a single reference sentence $\tilde{y}$ and using its masked contexts as a shared conditioning base, we obtain a consistent scoring function that requires only T masked-LM evaluations for **all possible sentences of length T** .
>
> In fact, we can also list the top-1000 sentences under $\tilde{p}_(\tilde{y} = \text{I like dogs})$, and are given by the cartesian product $S \times V \times O$, where
>
> $S \coloneqq$['I', 'Just', 'They', 'We', 'just', 'People', ' I', 'Looks', 'You', 'Sounds']
>
> $V \coloneqq$[' love', ' like', ' hate', ' have', ' want', ' eat', ' own', ' know', ' dislike', ' miss']
>
> $O \coloneqq$[' it', ' you', ' that', ' this', ' her', ' them', ' him', ' cats', ' music', ' dogs']
>
> We have also updated the submission to make this clearer.
>
> > “the introduction [of embeddings] of needs to be gentler”
>
> Please see our revised subsection 4.2

---

> ### Author Response · Authors · 2025-11-23
> **response 2 to review**
>
> > “Do we have any theoretical guarantees on how good this approximation is?”
>
> Please see section H in the appendix. In short, the error  depends on two quantities: (i) the spread of the embedding distribution and (ii) the mismatch between the anchor and the mean embedding. Under the locally contextualized distribution we can expect both of these to be small since sentences dissimilar to the anchor, or reference, are conditioned on the reference context, and are therefore very improbable under the locally contextualized distribution at that reference.
>
> > “We need to better motivate the path from knowledge compilation to average embeddings here… Relatedly, the discussion on how to compute the expected embedding (l343) reads as  disconnected to the previous part… Finally, a connection is missing from the last part of that subsection to the circuit being smooth and decomposable”
>
> Please see the revised section 4.3. We first describe the structural conditions that a computation graph must satisfy for the expected embedding computation to be tractable. We then show that, by construction, the computational graph for the expected embedding w.r.t. to our locally contextualized distribution abides by such structural constraints. Finally, we show how that yields a very simple, and efficient computation: a batched matrix multiplication.
>
> > “concrete suggestions to smooth the writing”
>
> Thank you for your valuable suggestions. We have implemented them all in the paper.

---

### Official Review · Reviewer_yLkE · 2025-11-01

**Soundness:** 2
**Presentation:** 3
**Contribution:** 2
**Rating:** 2
**Confidence:** 3

**Summary:**

Here is my understanding of the algorithm (please correct if wrong).
For each position,

1. Get AR logits, which are then used to get top-k tokens. Use these as “seeds” and expand the batch using this.

2. Using a short gibbs sampling chain to obtain look ahead completions, given the initial top-k tokens.

3. Use the MLM to obtain pseudo-logits, which are then used to compute an expected embedding.

4. Use the expected embeddings to compute the gradient of the constraint and use a first order taylor approximation to estimate the change in the constraint for each continuation.

They demonstrate superior empirical performance on sentiment control, language detoxification, and topic control.

**Strengths:**

**Empirical Performance**: The empirical performance does appear quite strong, and the paper demonstrates that this does scale to larger models.

**Experiment Design**: The experimental design also seems very strong: the selected prompts for each task (toxicity, sentiment, topic) seem appropriate given the constraint; and the evaluation metrics are reasonable.

**Weaknesses:**

**Framing and Motivation**: The paper is framed as performing exact inference of an approximate distribution, but I feel that this is not quite accurate. Finite time Gibbs is an MCMC method, which converges as the number of sampling steps -> infinity. Pseudo-likelihoods from a masked language model do not necessarily correspond to a valid joint distribution [1]. Thus the gibbs sampling algorithm may not even be sampling from a well defined distribution. The Taylor approximation also introduces another approximation layer.

As a result, I do not find the overall motivation for this approach to be convincing.

**Complexity**: The method is extremely complex for an algorithm sampling from an approximation of the true distribution — it requires 3 models (masked language model, autoregressive base, and external classifier). This isn’t a negative in itself: plenty of effective algorithms for sampling (no U-turn sampler) are complex.

The issue is that this complexity seems at odds with the motivation of using an approximate distribution instead of the actual distribution: typically an approximate distribution is used because it enables an easier target to sample from. In this case, their approximate distribution still necessitates the use of an external MLM to estimate pseudo-likelihoods, on top of first order Taylor approximations and computational circuits.

**Incomplete Literature Review**: [2] was published at ICLR last year and also focuses on semantic control via gradients. They guide an autoregressive model to satisfy external, differentiable constraints using biases computed from the gradient of the constraint. They rely on discrete gradient based sampling, which can also be interpreted as applying a Taylor approximation to the log density of the target distribution (in this case, the constraint) [3]. They also perform experiments on sentiment control, language detoxification, and topic control. The experiments presented in SCoNE are arguably more thorough and larger scale (DAB focuses on GPT2 size models, but SCoNE examines performance on LLama), but it would at least be worth mentioning this work, discussing the divergent motivations, and how SCoNE compares. While this work definitely does include larger scale experiments and has a slightly different experimental setup (and I prefer SCONE's experimental design), it would be at least worth discussing.

[1] Exposing the Implicit Energy Networks Behind Masked Language Models via Metropolis-Hastings. Goyal et al. ICLR 2022.

[2] Controlled LLM Decoding via Discrete Auto-regressive Biasing. Pynadath, Zhang. ICLR 2025.

[3] A Langevin-like Sampler for Discrete Distributions. Zhang et al. ICML 2022.

**Questions:**

1. What is the actual benefit over “exact inference of approximate distribution” v.s “approximate inference with exact distribution”?


2. What is the motivation behind this method? What is the problem that this method identifies with prior approaches, and how does it fix this?  It is mentioned that SMC deals require a lot of samples, and BON doesn’t factor in constraints during generation, but EBM based approaches do both. What is the insight that I should take away from reading this paper regarding controllable generation?


3. How are incompatible vocabularies handled? Does ModernBERT MLM / classifier use the same tokenizer as Llama?


4. What are the perplexities when sampling the base models using top-k of 10, which is the default for SCoNE? If scone uses a top-k of 10, then the natural baseline perplexity would be when sampling the AR model using top-k of 10.


5. (This is not a critique, I am asking purely out of interest) If the algorithm design hinges on having access to marginals for each position, which requires bidirectional attention, why choose a base model to be autoregressive? It feels that this algorithm would be much more natural to apply on masked diffusion language models, like LLada or Dream? These would circumvent the problem with the autoregressive factorization, remove the need for computing expectations using a different model's probability, and perhaps simplify some of the algorithm. How would the algorithm need to change — would it increase or decrease the complexity?

---

> ### Author Response · Authors · 2025-11-23
> **Response 1 to review**
>
> We would like to thank the reviewer for their thorough review.
>
> > *The paper is framed as performing exact inference of an approximate distribution, but I feel that this is not quite accurate… The Taylor approximation also introduces another approximation layer*
>
> The idea is that, by defining a simplified model whose dependency structure factorizes *nicely*, we can reason about *all* instances, or sentences, in the support of that simplified model. One instance of this would be assuming a fully-factorized, or *mean-field* model, essentially a unigram where we query the model for the probability of each token and maintain a table of size $\mathcal{V}$ in memory. **Such a model admits tractable and efficient expectations of simple functions** i.e. we can compute $\mathbb{E}[f]$ for a linear function $f$. However, as we know, **unigrams make for very bad language models**. So then the question becomes, can we come up with another distribution that remains tractable for $\mathbb{E}[f]$ while capturing richer linguistic structure. Therefore, we define what is essentially a *contextualized mean-field model*. Under the aforementioned model, the $i$th token of every new sentence $\mathbf{y}$ is conditioned on a reference sentence $\mathbf{\tilde{y}}$. This induces a distribution where sentences semantically similar to $\mathbf{\tilde{y}}$ are assigned high probability while those dissimilar are assigned low probability. Under such a distribution, we can still compute $\mathbb{E}[f]$ exactly while maintaining linguistic integrity.
>
> Unfortunately, even when $f$ is a very simple neural network, such tractability guarantees break.
> Approximating the verifier via a first-order Taylor expansion around the reference sample neatly reduces the problem back to taking an expectation of a linear function, restoring our tractability guarantees. This first-order Taylor approximation is justified under two assumptions. First, the contextualized distribution over generated samples is local, meaning most of the probability mass is concentrated around the model’s most likely completions and their semantically similar variants. Second, semantically similar sentences tend to have similar embeddings, which in turn implies similar outputs from the verifier function (assuming the verifier is reasonably smooth in embedding space). Together, these assumptions suggest that the verifier function varies smoothly and approximately linearly within this local region, making a first-order approximation a reasonable choice.
>
> The above approximations define a simplified model inside which we can perform exact inference. This fits neatly into the classical paradigm of exact inference in an approximate model (Koller & Friedman, 2009) where we approximate the distribution as well as the energy function.
>
> > *may not even be sampling from a well defined distribution*
>
> Although the conditional distributions produced by ModernBERT are not guaranteed to correspond to any globally consistent joint distribution, the resulting Gibbs update rule still defines a valid Markov chain on a finite state space. Under mild assumptions, this chain admits a unique stationary distribution and therefore converges to a well-defined equilibrium, even if no underlying MRF exists whose conditionals exactly match those of ModernBERT. Our goal is not to recover the true LM joint distribution but to use these locally normalized conditionals as an **efficient** proposal mechanism for estimating attribute probabilities within our local approximate model. We are okay trading slightly biased samples for more efficiency, which is not new in the probabilistic inference literature (e.g. the very famous Hogwild! Gibbs sampling is biased). Also note that strictly speaking our method does not require Gibbs sampling: the approximate conditional marginals can also be computed analytically as in [1] instead of using ModernBert to approximate them. We can even use alternative sampling strategies including vanilla Monte Carlo to obtain the reference/seed samples. Again, it’s a matter of trading efficiency and bias.
>
> Here are some examples of continuations obtained using this approach:
> - "I can't believe it, she's so very naughty."
> - "I can't believe it, she's so much to me."
> - "I can't believe it, she's so dashing hot!"
> - "I can't believe it, she's so sad."
> - "I can't believe it, she's so small and white."
>
> - "I can't believe it, he's so good at it!"
> - "I can't believe it, he's so good to her!!"
> - “I can't believe it, he's so lovin'!”
> - "I can't believe it, he's so completely above it."
> - "I can't believe it, he's so in tune... :)"

---

> ### Author Response · Authors · 2025-11-23
> **Response 2 to review**
>
> > "The method is extremely complex"
>
> We can analytically compute the approximate marginal conditionals alleviating the need for a masked LM. However, masked LMs provide am efficient high-quality approximation of the conditional marginals. Using other probabilistic models in tandem with a target model has been done before with great success [2][3], while also being the cornerstone of speculative decoding. In the former case it models “missing” probabilistic interactions, while in the latter it allows for more efficient computations.
>
> We would also like to point out that we never need to “materialize” a circuit. Rather, we draw a parallel between how our proposed approach models the structure of a circuit, thereby providing tractability guarantees. Computing the expected embedding reduces to performing a very efficient batched matrix multiplication. We have updated Section 4.3 in the paper to make this clear.
>
> > “complexity seems at odds with the motivation”
>
> We wish to sample from the distribution $p(Y_t | a, y_{<t}) \propto p(Y_t |, y_{<t}) \cdot p(a | y_{<t}) $. Computing the second term $p(a | y_{<t})$ is computationally hard: it would require us to evaluate the verifier on *every* continuation of $y_{<t}$. Note that there is not “easy” way of sampling from p(Y_t | a, y_{<t}). Therefore, we aim to estimate the term $p(a | y_{<t})$. The easiest way of doing that would be through Monte Carlo sampling: this is an unbiased, yet very high variance estimator of the estimated attribute probability, not to mention the expensive LLM sampling. High variance estimators are very detrimental for both learning and inference: that is precisely why we prefer the biased Gumbel-softmax estimator over the unbiased REINFORCE estimator, for instance.
>
> > "Incomplete Literature Review"
>
> We have added a comparison between the two approaches in Section 6 and Appendix F.
>
> > "What is the actual benefit over “exact inference of approximate distribution” v.s “approximate inference with exact distribution”?"
>
> Exact inference in an approximate model (our approach) trades a small bias for dramatically lower variance. Approximate inference with the true LM distribution, e.g., Monte Carlo sampling, can in principle yield an unbiased estimate, but in practice suffers from high variance and requires tens to hundreds of samples to obtain a stable estimate of an attribute expectation. In contrast, once a local approximate distribution is constructed, our estimator computes the expectation exactly with zero variance and at a fraction of the computational cost. This is the same rationale behind classical variational inference (Koller & Friedman, 2009): simplify the model so that exact inference becomes tractable, rather than keep the model exact and perform noisy inference. This is not unlike using the biased, low-variance Gumbel-softmax gradient estimator that performs exact inference in an approximate model in lieu of the unbiased, high-variance REINFORCE gradient estimator which performs approximate inference in an exact model.
>
> > "Motivation? Problem identified? Take aways?"
>
> We wish to sample from the distribution $p(Y_t | a, y_{<t}) \propto p(Y_t |, y_{<t}) \cdot p(a | y_{<t}) $. Computing the second term $p(a | y_{<t})$ is computationally hard: it would require us to evaluate the verifier on *every* continuation of $y_{<t}$. Note that there is not “easy” way of sampling from p(Y_t | a, y_{<t}). Therefore, we aim to estimate the term $p(a | y_{<t})$.
>
> We define two **approximate but tractable experts**. Our locally contextualized distribution fills in for the distribution over sentences and the first-order Taylor approximation of the verifier fills in for the scoring expert which assigns a score to every sentence. We show that computing this product of experts in this approximate model reduces to performing simple algebraic manipulations of the **expected embedding wrt the approximate distribution**. The main computational hurdle is then computing the expected embedding **over all sentences** wrt the approximate distribution. It turns out that computing this quantity simply corresponds to a very simple and efficient (batched) matrix multiplication.
>
> > "Vocabulary Incompatibility"
>
> Indeed ModernBERT does not use the same tokenizer as Llama. We have included the details of this conversion in Appendix A.3.
>
> Q4. "Perplexities of top-k?"
>
> Thank you for the suggestion, we have added a comparison to top-k in all the tables.
>
> > "Choice of autoregressive?"
>
> Our choice of autoregressive models is purely due to their SoTA performance.
>
> ### References:
>
> [1] Kareem Ahmed, Kai-Wei Chang, and Guy Van den Broeck. A Pseudo-Semantic Loss for Autoregressive Models with Logical Constraints. In NeurIPS 2023.
>
> [2] Emile van Krieken, Pasquale Minervini, Edoardo Ponti, and Antonio Vergari. Neurosymbolic Diffusion Models. In NeurIPS 2025.
>
> [3] Anji Liu, Oliver Broadrick, Mathias Niepert, and Guy Van den Broeck.Discrete Copula Diffusion. In ICLR 2025.

---

### Author Response · Authors · 2025-11-23
**General Comment**

We would like thank the reviewers for their thoughtful, thorough, and constructive feedback

We are grateful that the strengths of the work were independently highlighted across multiple dimensions. Reviewers noted that SCoNE offers a lightweight, inference-time semantic control method that differs substantially from prior sampling, heuristic, or training-based approaches, while remaining practical and scalable to large LMs. They also emphasized that the method demonstrates strong empirical performance across toxicity, sentiment, and topic control without degrading fluency or diversity. Several reviewers further highlighted the breadth and rigor of the experimental design, including the use of multiple model families, diverse baselines, and multifaceted evaluation metrics. Finally, we appreciate the recognition of the paper’s solid theoretical justification, and relevance to the broader problem of controllable generation.

We thank the reviewers again for their careful assessment and for acknowledging the contributions and significance of this work. We will now move on to address the concerns addressed by each reviewer separately, referencing the revised submission throughout.

---

### Meta-Review · Area_Chair_Rg1d · 2025-12-30

**Summary:**

Most reviewers recommend rejection, primarily because they remain unconvinced by the paper’s core methodology and empirical justification, and they view the claimed novelty as insufficiently established despite some strong empirical results and clarifications in the rebuttal. The evaluation and inference procedure are too naive. As a result, the paper’s main conclusion is currently too uncertain to warrant acceptance at ICLR.

**Reviewer Concerns:**

The rebuttal addressed several presentation issues and added a few ablation analyses, which likely resolve some concerns about evaluation completeness. However, reviewers are still not persuaded that the exact inference framework is sound, given the chain of approximations. There are also several questions about robustness (generalization) that remain unsolved.

**Reviewer Scores:**

I expect the strong rejects would remain essentially unchanged, as the rebuttal is not strong enoug,h and most analyses are missing. Overall, the score spread would persist, with the majority still landing on reject.

---

### Decision · Program_Chairs · 2026-01-26

Reject